# Altitudinal variation in rhizosphere microbial communities of the endangered plant *Lilium tsingtauense* and the environmental factors driving this variation

Boda Liu,[1] Jinming Yang,[1] Wanpei Lu,[1] Hai Wang,[1] Xuebin Song,[1] Shaobo Yu,[1] Qingchao Liu,[1] Yingkun Sun,[1] Xinqiang Jiang[1]

**ABSTRACT** The rhizosphere soil properties and microbial communities of *Lilium tsingtauense*, an endangered wild plant, have not been examined in previous studies. Here, we characterized spatial variation in soil properties and microbial communities in the rhizosphere of *L. tsingtauense*. We measured the abundance of *L. tsingtauense* at different altitudes and collected rhizosphere and bulk soils at three representative altitudes. The results showed that *L. tsingtauense* was more abundant, and the rhizosphere soil was richer in nitrogen, phosphorus, potassium, water content, and organic matter and more acidic at high altitudes than at lower altitudes. The diversity and richness of rhizosphere bacteria and fungi increased with altitude and were higher in rhizosphere soil than in bulk soil. In addition, ectomycorrhizal fungi, endophytic fungi, and nitrogen-fixing bacteria were more abundant, and plant-pathogenic fungi were less abundant at high altitudes. Co-occurrence network analysis identified four key phyla (Bacteroidota, Proteobacteria, Ascomycota, and Basidiomycota) in the microbial communities. We identified a series of microbial taxa (Acidobacteriales, Xanthobacteraceae, and Chaetomiaceae) and rhizosphere soil metabolites (phosphatidylcholine and phosphatidylserine) that are crucial for the survival of *L. tsingtauense*. Correlation analysis and random forest analysis showed that some environmental factors were closely related to the rhizosphere soil microbial community and played an important role in predicting the distribution and growth status of *L. tsingtauense*. In sum, the results of this study revealed altitudinal variation in the rhizosphere microbial communities of *L. tsingtauense* and the factors driving this variation. Our findings also have implications for habitat restoration and the conservation of this species.

**IMPORTANCE** Our study highlighted the importance of the rhizosphere microbial community of the endangered plant *L. tsingtauense*. We found that soil pH plays an important role in the survival of *L. tsingtauense*. Our results demonstrated that a series of microbial taxa (Acidobacteriales, Xanthobacteraceae, Aspergillaceae, and Chaetomiaceae) and soil metabolites (phosphatidylcholine and phosphatidylserine) could be essential indicators for *L. tsingtauense* habitat. We also found that some environmental factors play an important role in shaping rhizosphere microbial community structure. Collectively, these results provided new insights into the altitudinal distribution of *L. tsingtauense* and highlight the importance of microbial communities in their growth.

**KEYWORDS** *Lilium tsingtauense*, rhizosphere environment, microbial communities, soil properties, soil metabolites

B iodiversity loss poses a significant challenge to society because species extinctions can trigger ecological collapse (1) and compromise the functioning of ecosystems

Address correspondence to Xinqiang Jiang, jiangxinqiang8@163.com, or Yingkun Sun, 200001033@qau.edu.cn.

The authors declare no conflict of interest.

vital for human societies (2). Consequently, the protection of threatened biodiversity and the restoration of degraded ecosystems have become urgent global priorities (3).

Soil microorganisms are the basis for the sustainable survival of endangered plant populations (4). However, the symbiotic relationships between endangered plants and soil microorganisms have been poorly evaluated. For example, *Bacillus subtilis* and *Pseudomonas fluorescens* are known to play key roles in protecting the endangered *Pulsatilla tongkangensis* (5). Certain fungi, such as *Trichoderma*, *Mortierella*, and *Hypocrea*, are thought to contribute to the protection of the natural habitat of the first-class endangered plant *Cypripedium japonicum* and to facilitate its reproduction (6). Consequently, studies of soil bacteria and fungi can provide valuable insights into the growth and survival of rare and endangered plants, their environmental stress resistance, and strategies that could be employed to ensure their long-term persistence.

Soil physicochemical properties and soil metabolites are important components of the soil environment. The soil environment, which provides important microbial habitats, can have significant effects on microbial community structure and diversity (7). For example, soil pH is considered a key predictor of soil microbial community composition and diversity (8). The addition of N and P to soil has been shown to increase the ratio of bacterial to fungal phospholipid fatty acids and alter microbial diversity (9). Soil metabolites are essential for the metabolic activities of microbes, including their catabolic activity and functional diversity (10). Additionally, interactions among microorganisms have been shown to modify the composition of diverse communities (11). Studies on the effect of the soil environment on microbial communities are thus critically important, as they can help clarify changes in the properties of microbial communities, predict future changes in community composition, and reveal the mechanisms driving microbial community assembly.

*Lilium tsingtauense* (Liliaceae) is a rare wild lily (12) that primarily occurs on shaded or partially shaded slopes of Laoshan Mountain in Shandong Province, China, at altitudes (ALTs) ranging from 200 to 1,000 m. This species is also an important floral germplasm resource because of its distinctive verticillate leaves and strong cold resistance. Many natural populations of this species have become extirpated due to fragmentation or degradation of its habitat. *L. tsingtauense* has been designated as a nationally protected wild plant in China (13). More studies of this endangered plant are needed to ensure the long-term persistence of its natural populations.

In this study, we measured the abundance of wild *L. tsingtauense* at different altitudes and analyzed rhizosphere soil (RS) properties at three representative altitudes (low, medium, and high). Next, we compared the rhizosphere microbiome and the bulk soil (BS) microbiome in samples collected at the same altitude. We also analyzed the effects of some environmental factors on the rhizosphere microbiome of *L. tsingtauense* using correlation analysis and random forest analysis. This is the first large-scale investigation of the rhizosphere microbial community of *L. tsingtauense*, which provides an important value for exploring the characteristics of its rhizosphere microorganisms. These findings have implications for the conservation of endangered plants and their adaptation to future environmental change.

## RESULTS

### *L. tsingtauense* is more abundant in high-altitude areas

According to the characteristics of *L. tsingtauense* community, we selected three types of soil sampling sites [high altitude (HA), medium altitude (MA), and low altitude (LA)] and recorded them (Fig. 1A and B). We found that the abundance of wild *L. tsingtauense* at 800–1,000 m (HA) was greater than that at other altitudes LA and MA (Fig. 1C). Plant height was higher at HA than at LA and MA (Fig. 1D). We also found that *L. tsingtauense* was most abundant on northwestern and southwestern slopes of 5°–35° (Fig. 1E and F). These results suggest that high altitude and shady slope direction may be more suitable for the growth of *L. tsingtauense*.

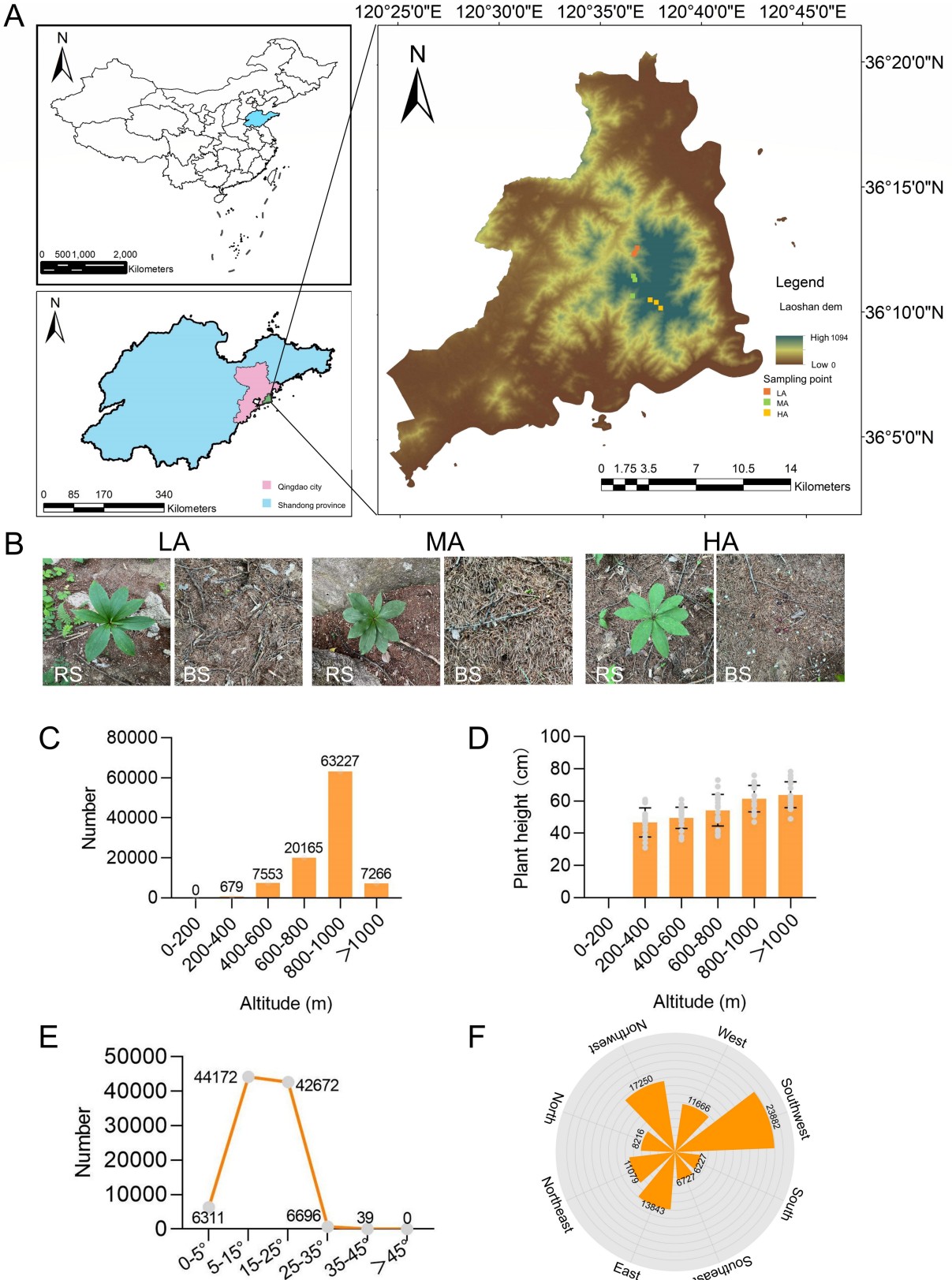

**FIG 1** Characteristics of sampling sites and *L. tsingtauense* community. (A) Detailed location of sampling points on the map. (B) BS and RS sampling points at LA, MA, and HA. The abundance (C) and plant height (D) of *L. tsingtauense* at different altitudes. The abundance of *L. tsingtauense* under different slopes (E) and slope orientations (F). Shaded slopes: north, northwest, west, southwest, and south; sunny slopes: north, northeast, east, southeast, and south. Bars represent standard errors (*n* = 20). BS, bulk soil; HA, high altitude; LA, low altitude; MA, middle altitude; RS, rhizosphere soil.

## Soil properties of bulk soil and rhizosphere of *L. tsingtauense* at different altitudes

We next analyzed the soil properties of *L. tsingtauense* of the RS and BS at three different altitudes (LA, MA, and HA) (Fig. 2). The water content (WC) and content of total N (TN), total P (TP), total potassium (TK), and organic matter (OM) were higher and the pH was lower in RS than in BS ($P < 0.05$). Additionally, we found that WC and TK increased with altitude in rhizosphere, and the pH decreased with increasing altitude. OM and TN initially increased and then decreased with altitude, and TP remained relatively stable with changes in altitude. These results indicated that RS accumulated more organic components than BS, and the soil properties of the rhizosphere are responsive to variation in altitude.

## Richness and diversity of bulk soil and rhizosphere microbial communities

To clarify the diversity and composition of the microbial community in *L. tsingtauense* at different altitudes, we sequenced the amplicons of the bacterial 16S rRNA gene and fungal ITS region from 18 soil samples after mass filtration. Following quality control and merging of paired-end reads, a total of 1,428,344 clean reads for bacteria and 1,240,367 clean reads for fungi were obtained. Each sample yielded at least 57,206 and 58,562 clean reads, with an average of 79,352 and 68,909 clean reads for bacteria and fungi, respectively (Tables S1 and S2).

Rarefaction curves, Shannon index curves, and species accumulation curves gradually plateaued, indicating that the sample sequences were sufficient for subsequent data analysis (Fig. S1 through S3). To clarify the microbial communities of both BS and RS, we analyzed the community composition of bacteria and fungi. Proteobacteria and Acidobacteria were the dominant bacterial phyla in BS and RS (Fig. 3A), and Ascomycota and Basidiomycota were the dominant fungal phyla in BS and RS (Fig. 3B).

We also evaluated the community richness and diversity of *L. tsingtauense* rhizosphere bacteria and fungi using the ACE, Chao1, Shannon, and Simpson indexes. The results showed that the richness and diversity of fungi and bacteria increased with altitude in RS (Fig. 4A and B). The bacterial richness and diversity were higher at HA in BS,

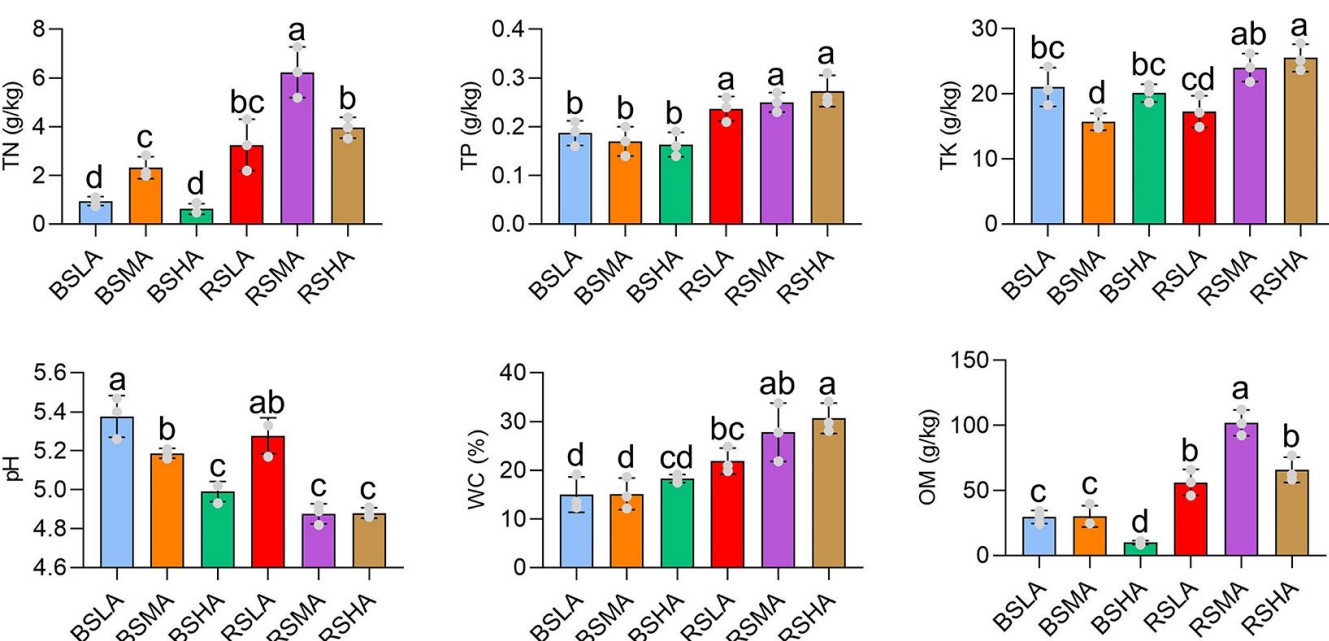

**FIG 2** Bar plot showing the properties of RS and BS at different altitudes. BS, bulk soil; HA, high altitude; LA, low altitude; MA, middle altitude; OM, organic matter. RS, rhizosphere soil; TK, total potassium; TN, total nitrogen; TP, total phosphorus; WC, water content. Different letters indicate significant differences according to analysis of variance plus Tukey's multiple comparison test ($P < 0.05$). Bars represent standard errors ($n = 3$).

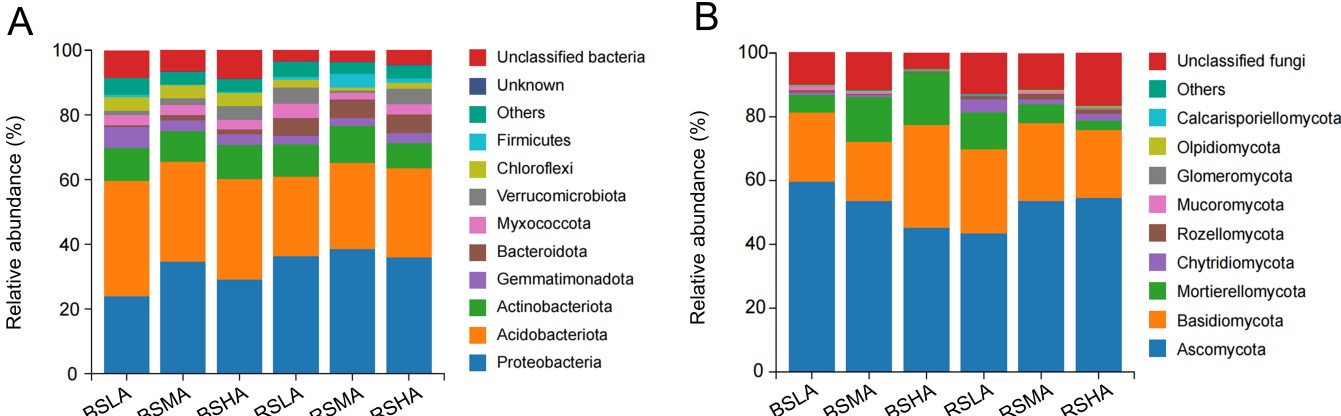

**FIG 3** Taxonomic distribution of bacterial and fungal communities. Bar plots show the composition of bacterial (A) and fungal (B) phyla. BS, bulk soil; HA, high altitude; LA, low altitude; MA, middle altitude; RS, rhizosphere soil.

while the fungal richness and diversity were lower. In addition, compared with BS, RS had more abundant and diverse bacterial and fungal communities at the same altitude (Fig. 4A and B). These findings indicate that the bacterial and fungal communities are related to altitude and RS of *L. tsingtauense*.

## Network associations and microbial hubs in rhizosphere bacterial and fungal communities

We next estimated the relationships among microbial taxa by constructing correlation networks for soil samples from RS and BS. The bulk soil bacterial network consisted of 281 nodes, including 3,440 positive edges and 3,179 negative edges (Fig. 5A and B). The rhizosphere bacterial network comprised 446 nodes, with 8,319 positive edges and 4,705 negative edges (Fig. 5C and D). In the fungal community, the bulk soil fungal network had 297 nodes, including 3,749 positive edges and 2,325 negative edges. The rhizosphere fungal network consisted of 382 nodes, with 4,241 positive edges and 2,442 negative edges (Fig. S4). Average degree, average pathway length, modularity, and positive correlation line were higher in the bacterial and fungal co-occurrence networks in RS than in BS. The negative correlation ratio between bacteria and fungi was lower in RS than in BS (Table 1).

Details regarding the bacteria and fungi classified as nodes in the connector, module hubs, and network hubs are provided in Tables S3 to S6. Nodes with Zi of ≥2.5 or Pi of ≥0.62 were identified as keystone species, which indicates that they play a key role within the co-occurrence networks. The important nodes in the rhizosphere bacterial and fungal networks are more specialized than those in the bulk soil bacterial and fungal networks. The significant nodes in the rhizosphere bacterial networks mainly comprised Bacteroidota (30.8%) and Proteobacteria (53.8%) (Fig. 5E and F); in the fungal community, the significant nodes mainly comprised Ascomycota (72.4%) and Basidiomycota (23.2%) (Fig. S5).

The co-occurrence patterns of soil microbial communities at different altitudes (LA, MA, and HA) were further explored. The modularity index in the six networks indicates that the generated network is modular. The links, nodes, and average degree of the network of bacteria and fungi at HA are higher, which indicates that there is a more complex relationship between soil bacteria and fungi at LA. At the same time, Bacteroidota, Proteobacteria, Ascomycota, and Basidiomycota were the most abundant at the three altitudes (LA, MA, and HA) and dominated in the network (Fig. S6).

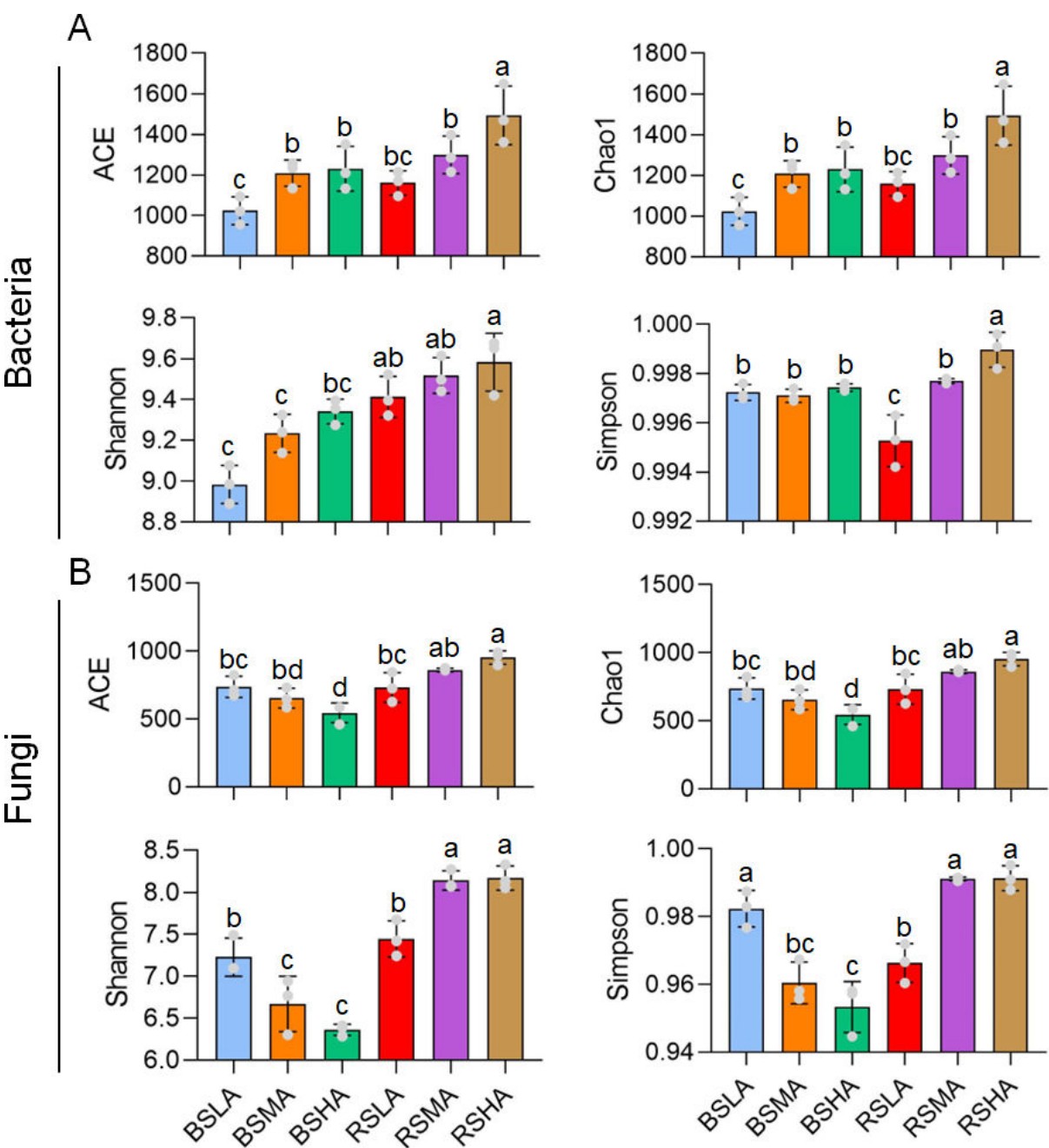

**FIG 4** Diversity indexes of various groups of rhizosphere bacterial and fungal communities. General patterns of bacterial (A) and fungal (B) alpha diversity along an altitudinal gradient, according to ACE, Chao1, Shannon, and Simpson indexes. Different letters indicate significant differences according to analysis of variance followed by Tukey's multiple comparison test ($P < 0.05$). Bars indicate standard errors ($n = 3$). BS, bulk soil; HA, high altitude; LA, low altitude; MA, middle altitude; RS, rhizosphere soil.

## Comparison of rhizosphere and bulk soil bacterial and fungal communities at different altitudes

To clarify the variability in bacterial and fungal taxa at different altitudes, we conducted an analysis of all ASVs at the family level and identified the five most dominant bacterial and fungal families. We found that Acidobacteriales and Xanthobacteraceae were the main rhizosphere bacterial families in the rhizosphere (Fig. 6A). The abundance of Acidobacteriales was relatively stable at different altitudes, and the abundance

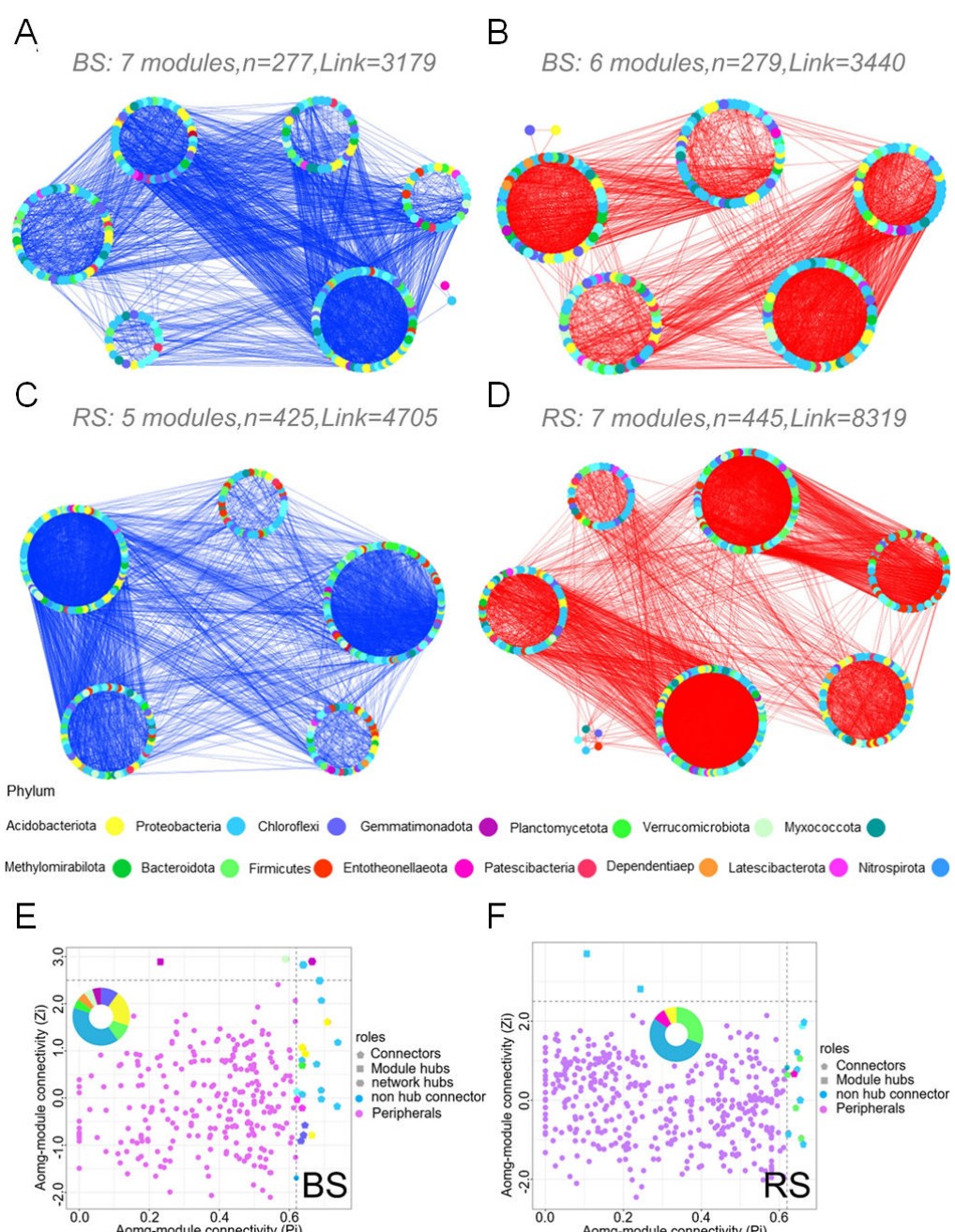

**FIG 5** Bacterial co-occurrence networks in RS and BS. Blue lines indicate negative correlations (A and C), and red lines indicate positive correlations (B and D); the co-occurrence network shows amplicon sequence variant (ASVs) with abundances greater than 0.01%. Zi–Pi plot showing the distribution of ASVs based on their topological roles in BS (E) and RS (F). All the nodes with $Zi \geq 2.5$ or $Pi \geq 0.62$ were keystone species, which corresponded to nodes in the area of connectors, module hubs, and network hubs that played a key role in the co-occurrence networks.

of Xanthobacteraceae increased significantly with altitude (Fig. 6A). Aspergillaceae and Chaetomiaceae were the main rhizosphere fungal families (Fig. 6B). The relative abundance of Chaetomiaceae was significantly higher at HA (7%) than at LA (5%) (Fig. 6B).

We also predicted the functions of bacteria and fungi in RS. We found that ectomycorrhizal fungi (28.7%), endophytic fungi (8.3%), and N-fixing bacteria (8.1%) were more

**TABLE 1** Topological characterization of bacterial and fungal co-occurrence networks in RS and BS of *L. tsingtauense*[a]

|          |     | NDN | EDN    | AD    | APL  | MOD  | NCL   | PCL   | NCR  |
|----------|-----|-----|--------|-------|------|------|-------|-------|------|
| Bacteria | BS  | 281 | 6,619  | 47.11 | 2.19 | 0.27 | 3,179 | 3,440 | 0.48 |
|          | RS  | 446 | 13,024 | 58.40 | 2.28 | 0.39 | 4,705 | 8,319 | 0.36 |
| Fungi    | BS  | 297 | 6,074  | 40.90 | 2.32 | 0.31 | 2,325 | 3,749 | 0.38 |
|          | RS  | 382 | 6,683  | 34.99 | 2.41 | 0.33 | 2,442 | 4,241 | 0.37 |

[a]AD, average degree; APL, average pathway length; EDN, edge number; MOD, modularity; NCL, negative correlation line; NCR, negative correlation ratio; NDN, node number; PCL, positive correlation line.

abundant, and plant-pathogenic fungi (7.9%) were less abundant in RS at HA than at LA (Fig. 7).

## Characteristics of metabolites in the RS of *L. tsingtauense* at different altitudes

To clarify the composition of RS metabolites in *L. tsingtauense*, we conducted liquid chromatography–mass spectrometry (LC–MS) non-targeted metabolomics analysis of soil samples from three altitudes. LC–MS detected a total of 4,188 peaks, of which 1,427 metabolites were successfully annotated. The Kyoto Encyclopedia of Genes and Genomes (KEGG) database was used to annotate all the identified metabolites, and the top 20 pathway with the most annotations in KO pathway level 2 was selected (Fig. 8A). These annotations revealed the enrichment of seven secondary metabolic pathways, which accounted for a high proportion of metabolic pathways, including terpenoid and polyketide metabolism (17.21%), chemical structure transformation maps (15.49%), and lipid metabolism (11.27%) (Fig. 8A).

We next analyzed the heat map of the 50 most abundant rhizosphere metabolites at different altitudes and found that the soil metabolites were similar at MA and HA (Fig. 8B). Volcanic maps indicated that there existed more phosphatidylcholine and phosphatidylserine in HA than in MA and LA (Fig. 8C through E).

## Effects of environmental factors on *L. tsingtauense* and rhizosphere microbial community

Spearman's correlation analysis revealed that phosphatidylcholine, 15-Hydroxyeicosatetraenoic acid (15(S)-HETE), anemarsaponin E, ALT, TK, pH, fungal average degree (FAD), and bacterial negative correlation ratio (BNR) were most closely related to the growth and rhizosphere microbial community of *L. tsingtauense* (Fig. 9). Additionally, phosphatidylcholine ($r$ = 0.72, 0.31, 0.87, and 0.64), 15(S)-HETE ($r$ = 0.68, 0.67, 0.63, and 0.82), ALT ($r$ = 0.82, 0.6, 0.83, and 0.88), and TK (0.72, 0.37, 0.75, and 0.79) increased the richness and diversity of rhizosphere bacteria and fungi, as well as the abundance and height of *L. tsingtauense*. Anemarsaponin E ($r$ = −0.71, −0.45, −0.68, and −0.80) and BNR ($r$ = -0.83, −0.57, −0.84, and −0.84) were significantly negatively correlated with the richness and diversity of rhizosphere bacterial and fungal communities, and this reduced the abundance and height of *L. tsingtauense*. Analysis of the pH indicated that *L. tsingtauense* was more suitable for acidic soil and played an important role in maintaining the diversity and richness of rhizosphere bacteria ($r$ = −0.68 and −0.6) and fungi ($r$ = −0.72 and −0.89). Furthermore, the random forest model showed that BNR, bacterial modularity, FAD, TK, and phosphatidylcholine had the greatest effect on the distribution and growth of *L. tsingtauense*. Anemarsaponin E, 15(S)-HETE, phosphatidylcholine, WC, and FAD contributed the most to explaining variation in the diversity and richness of bacterial and fungal communities in the rhizosphere of *L. tsingtauense*.

## DISCUSSION

The rhizosphere is a narrow and dynamic area of the plant root–soil interface. It is considered one of the most complex and functionally active ecosystems on the planet,

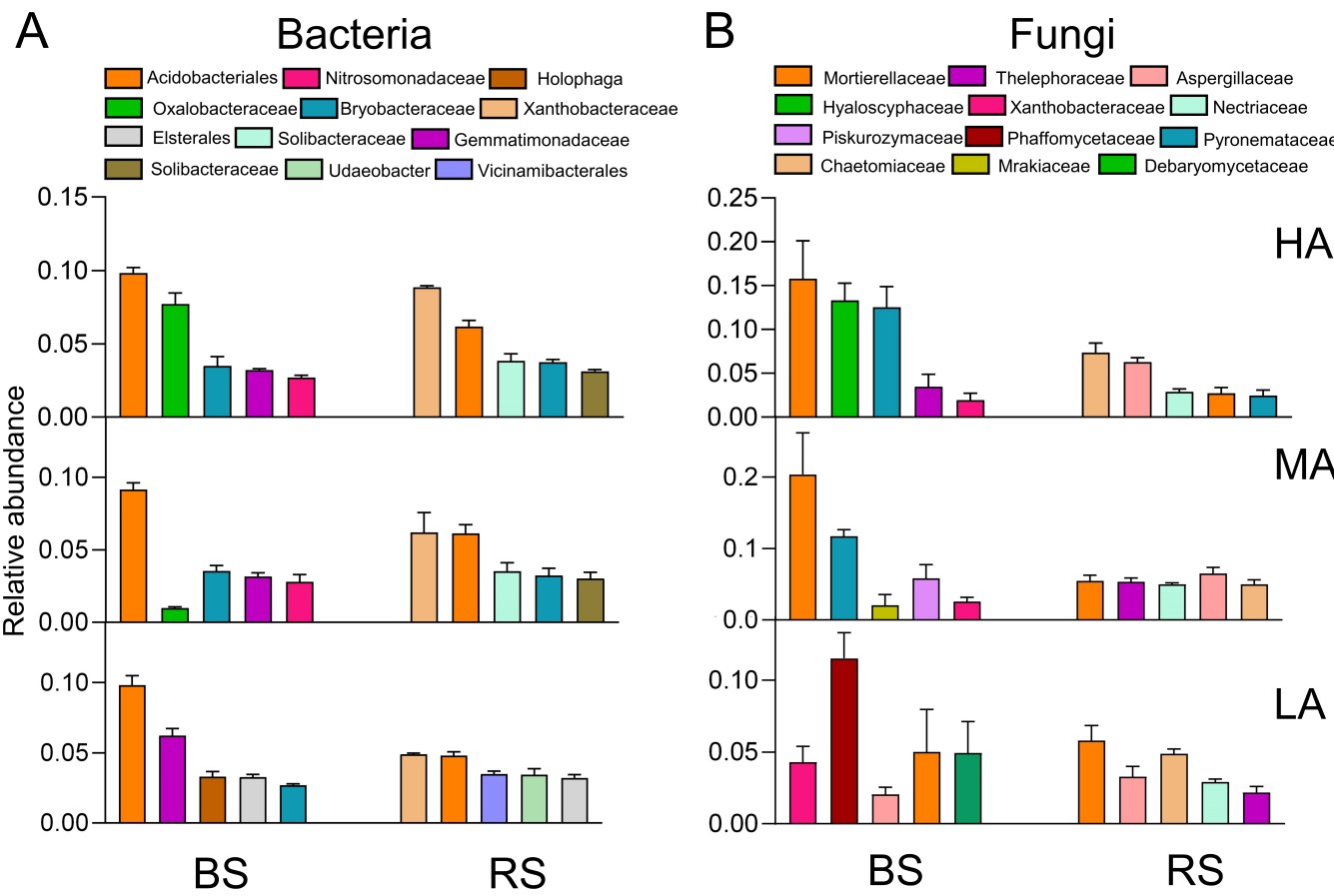

**FIG 6** The distribution of bacterial and fungal families in RS and BS at different altitudes. (A) Distribution of bacterial and fungal families (B) at LA, MA, and HA in BS and RS. Bars indicate standard errors ($n = 3$). BS, bulk soil; HA, high altitude; LA, low altitude; MA, middle altitude; RS, rhizosphere soil.

and it can promote plant health and mitigate the effects of biotic and abiotic stresses (14). Study of the composition and function of the rhizosphere microbiome is essential for clarifying how the microenvironment affects plant growth and predicting the effects of future environmental change. Here, the composition of the rhizosphere microbiome associated with the endangered plant *L. tsingtauense* was characterized, and the environmental factors affecting rhizosphere microbiome communities were identified.

## High-altitude areas might be the suitable soil environment for *L. tsingtauense*

Plant roots are in close contact with the soil, and materials are frequently exchanged between the roots and soil (15, 16). Soil is a key ecological factor that can regulate plant growth (17). Our findings are consistent with the results of previous studies that the WC is typically higher in HA areas (Fig. 2), and the water available to plants in HA barren soils can support the normal physiological activities of the roots (17). A specific soil pH range is required for the survival of many plants. Our results showed that the pH of the rhizosphere of *L. tsingtauense* ranged from 4.8 to 5.4 (Fig. 2). This observed pH difference may be linked to variation in root exudates near the roots (18).

Soil OM, TN, TP, and TK are critical nutrients essential for plant growth and development (19). The high TK content in HA areas aids the growth of *L. tsingtauense* by providing a source of essential nutrients. Additionally, the increased TK content in the rhizosphere can induce the development of thick outer walls in epidermal cells, reducing the occurrence of plant diseases (20). TP is a fundamental macronutrient necessary for plant growth and development (21). Given that plants consume soil P, the roots absorb P from the rhizosphere (22, 23), which increases the P content in the rhizosphere.

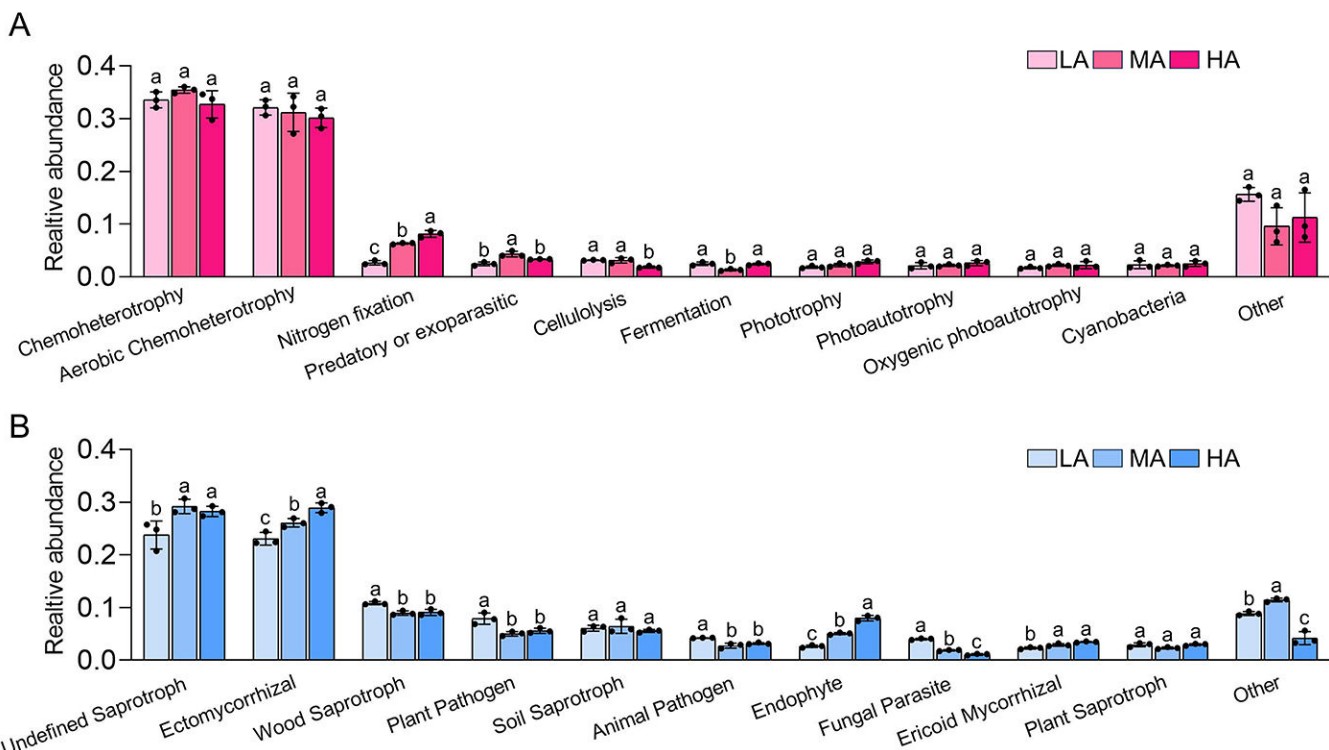

**FIG 7** Guild assignments for the obtained ASVs using the FUNGuild and FAPROTAX databases. The mean proportions of bacterial (A) and fungal (B) functions were calculated to estimate their relative abundances in the rhizosphere. Different letters indicate significant differences according to analysis of variance followed by Tukey's multiple comparison test ($P < 0.05$). Bars indicate standard errors ($n = 3$). HA, high altitude; LA, low altitude; MA, middle altitude.

## Characteristics and functions of bacterial and fungal communities in the rhizosphere of *L. tsingtauense*

Based on the composition of the rhizosphere microbial communities and co-occurrence networks, Proteobacteria, Acidobacteriota, Bacteroidota, Ascomycota, and Basidiomycetes were identified as the key phyla for *L. tsingtauense* growth (Fig. 3). These phyla likely make significant contributions to the composition and diversity of bacterial and fungal communities. Members of the Ascomycota are known to play a role in promoting plant development, resulting in regulated carbon and N cycling (23, 24). Basidiomycota species also play a role in the carbon cycle and ecosystem function (25). Proteobacteria are r-strategists that thrive in high-nutrient conditions (26) and on labile carbon sources (27). Bacteroidetes are abundant members of the plant microbiome with pathogen-suppressing abilities, and they promote the absorption and utilization of P in the rhizosphere (28). Acidobacteria mediate cellulose degradation (29) and are involved in various iron cycle-related and ecosystem-related functions (30, 31). These five taxa likely play key roles in shaping bacterial and fungal communities in the rhizosphere.

The richness and diversity of soil microorganisms were higher at all sites in RS than in BS (Fig. 4). The reason is likely that metabolites produced by plant roots recruit more rhizosphere microorganisms (32). The co-occurrence network is also consistent with these findings (Fig. 5; Fig. S4 and S5; Table 1); that is, the rhizosphere has a more complex and abundant microbial community. These complex microbial communities are conducive to plant protection and growth. For example, bacterial interactions lead to the production and detection of small chemical signaling molecules that induce systemic resistance, which protects plants from pathogens (14). Fungal interactions result in the formation of a common mycorrhizal network that facilitates nutrient transfer between plants and induces defensive responses against insect attacks (33).

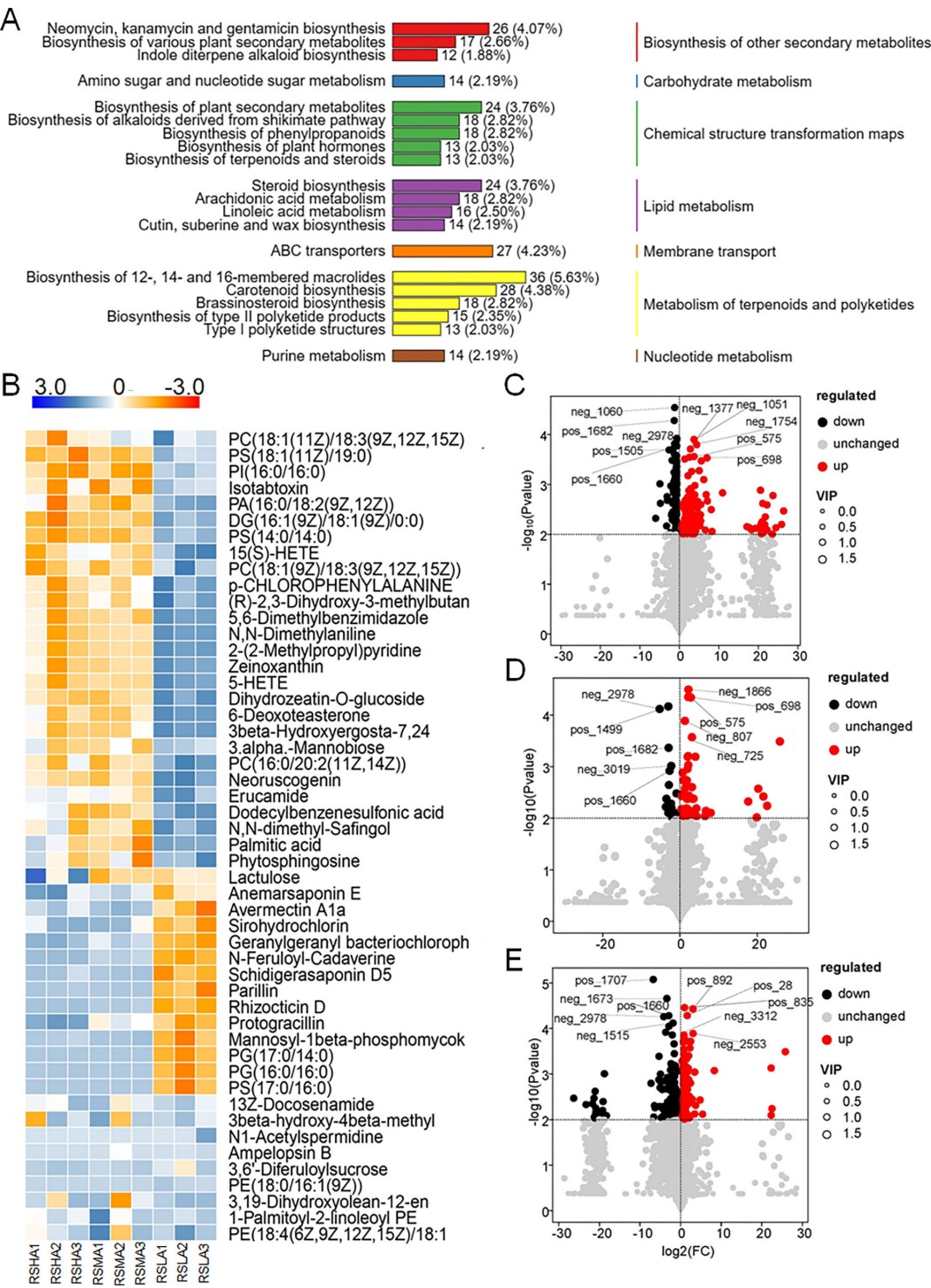

**FIG 8** Metabolic characteristics of the rhizosphere of *L. tsingtauense* at different altitudes. (A) The 20 KEGG pathways with the most annotations of RS metabolites. (B) Heat map analysis of the first 50 metabolites in RS at different altitudes. (C–E) Volcano maps of differential metabolites at different altitudes. The expression patterns of the 10 differential metabolites (five up-regulated and five down-regulated) with the lowest *P* value were shown. The names of the soil metabolites are shown in Table S7. BS, bulk soil; HA, high altitude; LA, low altitude; MA, middle altitude; RS, rhizosphere soil.

To identify the dominant bacterial and fungal species at a fine scale, we investigated changes in the responses of the microbiological community at different altitudes. We found that Acidobacteriales, Xanthobacteraceae, Aspergillaceae, and Chaetomiaceae are more abundant in RS than in BS (Fig. 6). Acidobacteriales can regulate the expression of

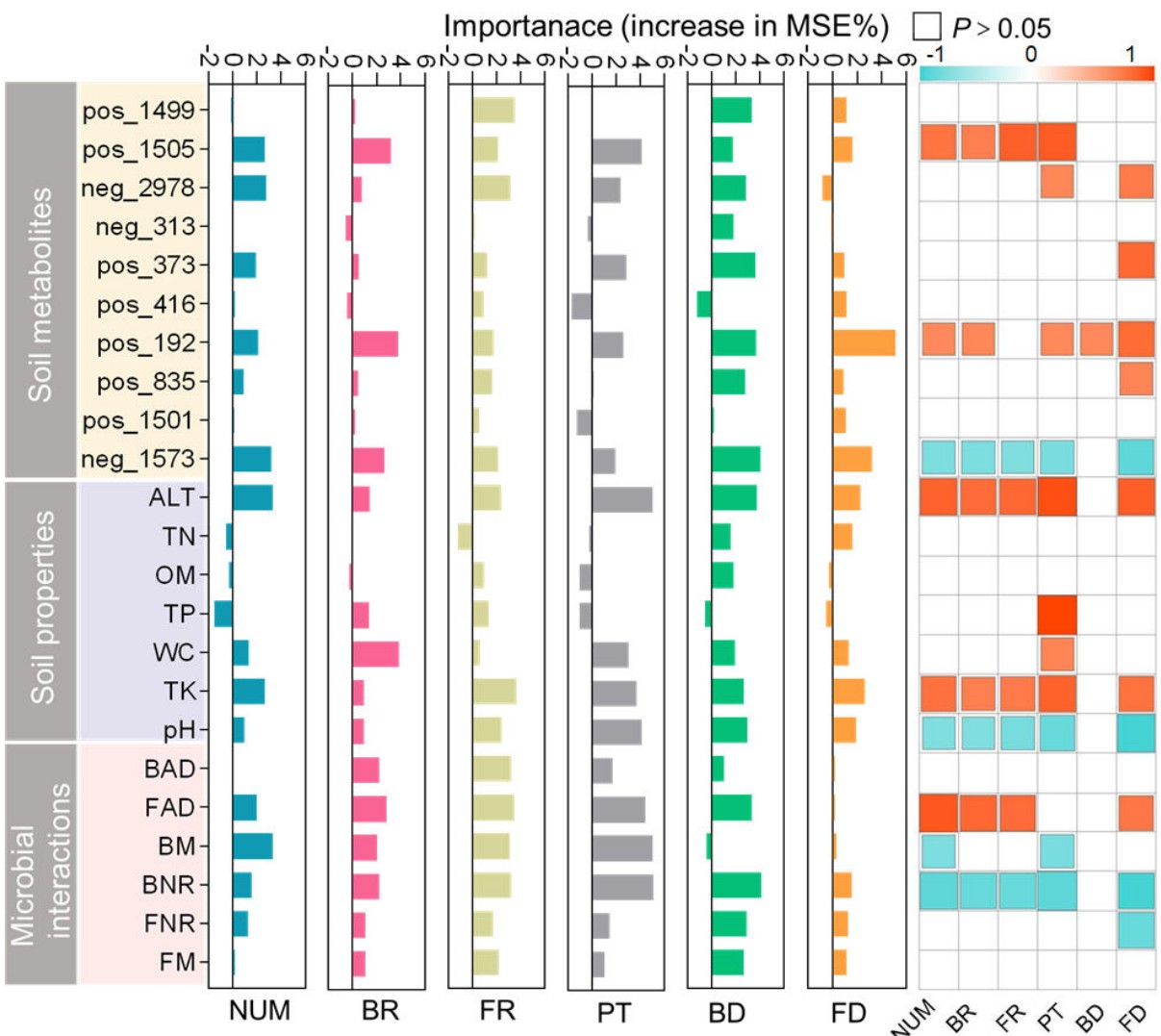

**FIG 9** The contributions of different environmental factors to the growth and rhizosphere microbial community of *L. tsingtauense* were analyzed based on correlation analysis and random forest analysis. The names of soil metabolites are shown in Table S7. Correlation analysis was performed using Spearman's correlation analysis ($P < 0.05$). Orange indicates $r > 0$, and light blue indicates $r < 0$. ALT, altitude; BAD, bacterial average degree; BD, bacterial diversity; BM, bacterial modularity; BNR, bacterial negative correlation ratio; BR, bacterial richness; FAD, fungal average degree; FD, fungal diversity; FM, fungal modularity; FNR, fungal negative correlation ratio; FR, bacterial richness; NUM, number; OM, organic matter; PT, plant height; TK, total potassium; TN, total nitrogen; TP, total phosphorus; WC, water content.

genes involved in N fixation, P dissolution, and auxin production in plants (34). Aspergillaceae is considered a potential biological control agent that has been shown to mitigate the deleterious effects of pathogens. Chaetomiaceae has been shown to be an endophytic fungus in some plants (35) that can increase the growth rate of host plants and improve their biotic stress tolerance (36). Furthermore, ectomycorrhizal fungi, endophytic fungi, and N-fixing bacteria were more abundant at HA than at LA (Fig. 7). N-fixing bacteria have N-fixing capabilities, and they can mediate the synthesis of plant growth substances both above and below ground (37), which can enhance the adaptability of most terrestrial plants. Ectomycorrhizal fungi and endophytic fungi are often engaged in biotic interactions with plants, including the nutritional relationship (38–40), which promotes seedling growth and nutrient uptake (41).

## Metabolic characteristics of *L. tsingtauense* rhizosphere

Plant–microbe communication is primarily achieved through chemical signaling (42). Studies of these chemical signals can enhance our understanding of the interactions between plants and rhizosphere microorganisms. We identified chemical signal molecules through analysis of rhizosphere metabolites. These small-molecule metabolites are primarily associated with terpenoid and polyketide metabolism as well as lipid metabolism pathways (Fig. 8A). They enhance the adaptability of plants to the environment (43). These metabolites can also enhance plant disease resistance, induce plant pathogen defenses, mediate the intricate interplay between plants and surrounding microorganisms, and improve the ability of plants to resist natural enemies (44).

To gain a deeper understanding of the metabolites in *L. tsingtauense* rhizosphere, we analyzed rhizosphere metabolic components of samples obtained at different altitudes. Our results showed that the metabolites of rhizosphere at MA and HA were similar, and the content of phosphatidylcholine and phosphatidylserine was higher at MA and HA than at LA (Fig. 8B through E). Phosphatidylcholine is the most abundant phospholipid in the eukaryotic cell membrane, and it plays an important role in signal transduction (45). For example, phosphatidic acid is one of the signal molecules produced by phosphatidylcholine (46). It has been shown to be involved in signaling pathways that regulate cell growth, proliferation, reproduction, and responses to hormones and biotic and abiotic stresses (47). Phosphatidylserine has been widely reported to promote plant growth and development and enhance resistance to abiotic stress (48). It has also been shown to activate plant auxin signals and regulate cell elongation and leaf senescence (49). These diverse rhizosphere soil metabolites indicate that they can enhance the survival of *L. tsingtauense*.

## Soil properties, microbial community, and soil metabolites are important for the distribution of *L. tsingtauense*

Previous studies have shown that high abundance of rhizosphere microbial communities can increase plant productivity (14). Therefore, identifying the environmental factors associated with rhizosphere microbial community richness can have implications for optimizing the microbial community of *L. tsingtauense* rhizosphere. The soil potassium content and pH enhanced the diversity and richness of the rhizosphere microbiome (Fig. 9). The amount of potassium absorbed by plants in the soil is small. A large number of microorganisms, especially rhizosphere microorganisms, have been shown to play a role in potassium dissolution and the transformation of potassium into plant-available forms (50). This attracts more microorganisms capable of dissolving potassium. When microorganisms dissolve potassium minerals, they often produce more organic acids to acidify the rhizosphere (51) and discharge them into the surrounding environment to reduce the pH (52). pH has been shown to regulate microbial communities extensively (53). Therefore, it enriches the rhizosphere microbial community and creates suitable conditions for the growth of *L. tsingtauense*.

Soil metabolites have a major effect on the structure and function of microbial communities (54). Some soil metabolites can enhance the growth and reproductive conditions for rhizosphere microorganisms (55) and inhibit the growth of other rhizosphere bacteria and fungi (56). However, the mechanisms underlying the effects of phosphatidylcholine, 15(S)-HETE, and anemarsaponin E on microbial communities remain unclear. We speculate that these soil metabolites affect plant–microbe interactions and alter the microbial communities in root soil.

Interactions between microorganisms have significant effects on the structure of microbial communities. In our study, the average degree of the fungal co-occurrence network and the negative correlation ratio of bacteria were closely related to the structure of rhizosphere bacterial and fungal communities (Fig. 9). The highly diverse fungal community can be used as nutrients available to the microbial community through rhizosphere metabolites, which positively regulate the richness of the rhizosphere microbial community (57). Antagonistic interactions among rhizosphere bacteria

can lead to the collapse of microbial communities (58), which can reduce diversity and richness.

## Conclusion

This study is the first report of a comprehensive reference for understanding the rhizosphere microbiome of *L. tsingtauense*. We analyzed the characteristics and functions of soil microbial communities in *L. tsingtauense* and determined the environmental factors affecting the differences in microbial communities. We also identified a series of microbial taxa, such as Acidobacteriales, Xanthobacteraceae, Aspergillaceae, and Chaetomiaceae, and rhizosphere soil metabolites, such as phosphatidylcholine and phosphatidylserine, which play an essential role in the survival of *L. tsingtauense*. In sum, these results highlight the importance of microbial communities for *L. tsingtauense* growth and provide profound insights into the driving factors of microbial communities in ecosystems. At the same time, our results also provide a new perspective view for the protection of endangered plants.

## MATERIALS AND METHODS

### Survey of *L. tsingtauense*

From March 2023 to November 2023, we investigated the wild distribution and characteristics of *L. tsingtauense* in Mount Lao, which is located in Qingdao, Shandong province of China (Fig. 1A). The growth slopes, slope directions, and number of *L. tsingtauense* at different altitudes were recorded, respectively. To assess the growth status of *L. tsingtauense* influenced at different altitudes, we randomly selected 20 plants and recorded their heights.

### Sampling area and sample collection

Three different representative altitudes, which contain the abundant wild *L. tsingtauense* community, were used as the soil sample sites. These sampling sites included low altitude (486 m, 120.6097°E, 36.2063°N; 516 m, 120.6105°E, 36.2071°N; and 524 m, 120.6124°E, 36.2103N°), middle altitude (688 m, 120.6082°E, 36.1786°N; 714 m, 120.6104°E, 36.1893°N; and 738 m, 120.6091°E, 36.1916°N), and high altitude (975 m, 120.6230°E, 36.1757°N; 985 m, 120.6314°E, 36.1702°N; and 1,029 m, 120.6277°E, 36.1739°N) and visualized by Arcmap (version 10.8). The sampling point of each altitude was in a 10 cm × 10 cm square area of three different sites, and the minimum distance between different sampling sites was above 50 m.

To characterize the composition and function of rhizosphere bacteria and fungi, we collected RS samples from a depth of 5–10 cm by gently shaking the plants to separate the soil from the roots. We also collected soil samples from areas without any plant growth at the same depth, which we defined as BS, and these were used as control samples. We finally collected 18 soil samples (3 RS + 3 BS × 3 different altitudes). These samples were divided into two groups. One group of soil samples was stored at −80°C for DNA extraction, and the other group of soil samples was sieved through a 2-mm sieve, dried, stored at room temperature, and then subjected to various assays to determine soil characteristics.

### Determination of soil characteristics

Soil properties, including soil pH and the content of OM, soil WC, TN, TP, and TK, were determined following the methods described by Bao (59).

### DNA extraction, PCR amplification, and sequencing

Total DNA of the rhizosphere microbiome of each sample was extracted using TGuide S96 magnetic bead extraction kits (TIANGEN, Beijing, China). The libraries of 16S

rDNA (bacteria) and internal transcribed spacer (ITS, fungi) were constructed by using the bacterial 16 S V3 + V4 primers (338F, 5′-ACTCCTACGGGAGGCAGCA-3′, and 806R, 5′-GGACTACHVGGGTWTCTAAT-3′) and the fungal ITS primers (ITS1F, 5′- CTTGGTCATT TAGAGGAAGTAA-3′, and ITS2R, 5′-GCTGCGTTCTTCATCGATGC-3′), respectively (60, 61). PCR amplification was performed in a total volume of 50 µL. The PCR amplification products were collected from a 2% agarose gel and purified using Vazyme VAHTSTM DNA Clean Beads (Vazyme Biotech, Nanjing, China). The Illumina Mi Seq platform (Biomarker Technologies Corporation, Beijing, China) was used to perform high-throughput sequencing analysis of bacterial and fungal rRNA genes in the purified mixed samples.

## Metabolite extraction

*L. tsingtauense* rhizosphere samples were suspended in 5 mL of methanol–acetonitrile–water solution (2:2:1, vol/vol/vol) and then ground using a grinder (60 Hz; Biospec, USA). Metabolites were separated using a Waters UPLC Acquity I-Class PLUS system (Waters, Shanghai, China) with Waters Acquity UPLC HSS T3 columns (1.8 µm, 2.1 × 100 mm) (Waters). The metabolites were then detected using the Waters UPLC Xevo G2-XS QTOF mass spectrometry system (Biomarker Technologies Corporation).

## Bioinformatics analyses

Trimmomatic (version 0.33) was used to filter the original data, and Cutadapt (version 1.9.1) (62) was used to identify and remove the primer sequences. Subsequently, USEARCH (version 10) (63) was used to splice the double-end reads, and UCHIME (version 8.1) was used to remove the chimeras (64). The high-quality sequences were used for subsequent analyses.

The DADA2 method in QIIME2 (version 2020.6) was used for quality control (65). ASVs were filtered by default using a threshold of 0.005% of the number of all sequences (66). The naive Bayesian classifier was used to classify the feature sequence using the UNITE (version 8.0) database and SILVA 138 database (67), and the species classification information corresponding to each feature was obtained. The community composition of each sample was determined at the phylum, class, order, family, genus, and species levels.

We used the Shannon, ACE, Chao1, and Simpson indexes to measure species richness and diversity in each sample. Functional analysis of soil bacteria and fungi was performed using the FUNGuild and FAPROTAX databases.

ASVs in samples with abundances of more than 0.01% were retained for network analysis to avoid possible biases. The co-occurrence network of microbial phylum levels at all altitudes and different altitudes ($P < 0.05$, $r > 0.6$) was constructed by Spearman's test. Network topology parameters were calculated, and the networks were visualized using Cytoscape (version 3.8.0) and Gpehi (version 0.9.2) (68). The Zi and Pi (Zi ≥ 2.5 or Pi ≥ 0.62) of each node in the microbial symbiotic network were calculated according to Guimerà and Deng (69, 70).

An analysis of variance followed by Tukey's multiple comparison test was performed to evaluate the significance. All correlation analyses were performed using Spearman's ($r > 0$ indicates positive correlation, $r < 0$ indicates negative correlation) rank correlation analysis. The soil metabolite data matrix was compared against KEGG data sets to determine the KEGG annotations of the metabolites. KEGG functional pathway analysis results were presented using the "cluster Profiler" and "reshape" R packages (71). The orthogonal partial least squares discriminant analysis (OPLS-DA) method was used to screen differential metabolites (DEMs). The corresponding OPLS-DA model was established to obtain the metabolite variable important in projection (VIP) values. Student's *t*-test and fold change analyses were used to assess differences in metabolites between the two groups (72). DEMs were identified using the following criteria: VIP >1 and $P < 0.05$. The number of up-regulated and down-regulated DEMs in each

treatment group was determined. Volcanic maps were used to clarify the distribution of DEMs at different altitudes.

Random forest analysis was used to identify the main environmental factors affecting bacterial and fungal communities using the "randomForest" package. The percentage increase in mean square error of the variables was utilized to analyze the contributions of the predictors.

## Statistical analysis

GraphPad Prism (version 8.0; Microsoft Windows, USA) software was used to generate bar graphs. Statistical analysis was performed using SPSS Statistics (version 22.0) software (IBM, USA). Significance levels were indicated using different letters ($P < 0.05$). Image layout was performed using Adobe Illustrator 2023 (Adobe, USA).

## ACKNOWLEDGMENTS

This work has received support from the Department of the Ecological Environment of Shandong Province and the Talent Introduction Program for Youth Innovation Team of Shandong Higher Learning (no. 018–1622001).

X.J.: conceptualization, visualization, and writing (original draft, review, and editing); Y.S.: conceptualization and visualization; B.L.: investigation, writing (original draft), data curation, formal analysis, and visualization; J.Y.: investigation, data curation, formal analysis, and visualization; W.L., H.W., X.S., Q.L., and S.Y.: investigation.

## AUTHOR AFFILIATION

[1]College of Landscape Architecture and Forestry, Qingdao Agricultural University, Qingdao, Shandong, China

## AUTHOR ORCIDs

Boda Liu http://orcid.org/0009-0006-3554-8676
Yingkun Sun http://orcid.org/0000-0002-0112-2101
Xinqiang Jiang http://orcid.org/0000-0003-0727-3354

## AUTHOR CONTRIBUTIONS

Boda Liu, Investigation, Visualization, Writing – original draft, Data curation, Formal analysis | Jinming Yang, Formal analysis, Investigation, Visualization, Data curation | Wanpei Lu, Investigation | Hai Wang, Investigation, Visualization | Xuebin Song, Investigation | Shaobo Yu, Investigation | Qingchao Liu, Investigation | Yingkun Sun, Conceptualization, Visualization | Xinqiang Jiang, Conceptualization, Visualization, Writing – original draft, Writing – review and editing

## DATA AVAILABILITY

The soil fungal data set was deposited in the National Center for Biotechnology Information Sequence Read Archive under accession no. PRJNA1014599, and the soil bacterial data set was deposited under accession no. PRJNA1014604.

## ADDITIONAL FILES

The following material is available online.

### Supplemental Material

**Supplemental figures (Spectrum00966-24-s0001.docx).** Fig. S1 to S6.
**Supplemental tables (Spectrum00966-24-s0002.xlsx).** Tables S1 to S7.

Open Peer Review

**PEER REVIEW HISTORY (review-history.pdf).** An accounting of the reviewer comments and feedback.

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
