## [Reviewer comments · Microbiology Spectrum]

Microbiology Spectrum

Altitudinal variation in rhizosphere microbial communities of the endangered plant *Lilium tsingtauense* and the environmental factors driving this variation

Boda Liu, Jinming Yang, Wanpei Lu, Hai Wang, Xuebin Song, Shaobo Yu, Qingchao Liu, Yingkun Sun, and Xinqiang Jiang

Corresponding Author(s): Xinqiang Jiang, Qingdao Agricultural University

Review Timeline:

Submission Date:	April 16, 2024
Editorial Decision:	July 1, 2024
Revision Received:	July 12, 2024
Accepted:	August 28, 2024

Editor: Xia Ding

Reviewer(s): Disclosure of reviewer identity is with reference to reviewer comments included in decision letter(s). The following individuals involved in review of your submission have agreed to reveal their identity: Massimiliano Cardinale (Reviewer #1)

Transaction Report:

DOI: <https://doi.org/10.1128/spectrum.00966-24>

Re: Spectrum00966-24 (Altitudinal variation in rhizosphere microbial communities of the endangered plant *Lilium tsingtauense* and the environmental factors driving this variation)

Dear Dr. Xinqiang Jiang:

Thank you for the privilege of reviewing your work. Below you will find my comments, instructions from the Spectrum editorial office, and the reviewer comments.

Revision Guidelines

Sincerely,
Xia Ding
Editor
Microbiology Spectrum

Reviewer #1 (Public repository details (Required)):

Accession number of bioproject was provided.

Reviewer #1 (Comments for the Author):

- L. 40-41 and 486-487: these are microbial taxa, not communities.
- L. 54-56: I do not agree with this statement. There are plenty of studies on the microbiome associated to endangered plant species. Either provide an opportune reference to this statement or remove it.
- L. 56-57: this sentence is incomplete. It has no meaning.
- L. 120-122: either indicate the name of the primers or add opportune references.
- L. 151-154: for the network analysis, please indicate: correlation method used, statistical threshold considered as significant ($p < 0.05$? or < 0.01 ? or < 0.001 ?), algorithm used for network organization.
- L. 179-180: replace "attitude" with "altitude".
- Fig. 1B is not cited in the text.
- L. 108 and L. 209: the statements about number of samples are discordant. Please, correct. I guess you have 18 samples and performed 36 sequencing analyses (one bacteria and one fungi per sample).
- Fig. 3: move "unclassified bacteria" and "unclassified fungi" to the top of the cumulative bars, next to "unknown".
- L. 249: write all these acronyms extensively the first time.
- Supplementary tables S3-S6: why only the phylum is reported for the nodes?
- L. 269-273: this information, currently placed in the legend of Fig. 5, must be reported in the materials and methods instead, and opportune references must be indicated to justify the choose of both Z_i and P_i thresholds.
- Fig. 7: why didn't you perform any statistics on these data? Why are these results not described anywhere in the text?
- L. 325-340: I suggest to indicate the name of the metabolites instead of their codes, which are rather uninformative.
- L. 350: I guess you mean Table S7 instead of S8.
- Conclusions: what you wrote is just a repetition of the study aim and findings. Try to re-write the conclusions highlighting the implications of the results and the future perspectives that they open.

Reviewer #2 (Public repository details (Required)):

The sequencing datasets have been deposited.

Reviewer #2 (Comments for the Author):

Lines 54-57. Please rewrite these two sentences. For the first one, I would suggest the authors change the word "understudied" to other words such as "poorly evaluated".

Lines 97-103. The methods to generate Fig 1 (a- f) data were missing. Can you mark your sampling sites in Fig 1a? I thought it is important to know where you collect your 3 bulk soil and 3 rhizosphere soil samples at each of the three altitude ranges (LA, MA, and HA). What criteria you used to select your sampling sites? Since you were examining altitudinal variation, how would you determine your sampling sites are good representation?

Lines 117-128. Please briefly describe how you generate the sequencing libraries for both bacterial and fungal communities. The methods described here are confusing. What Illumina platform you used, Hiseq 2500 or Miseq?

Line 135. I suspect the words "mass spectrometry" were missing after the QTOF?

Lines 152-153. Please rewrite the sentence. How about the negative correlations?

Line 177. As mentioned before, please provide the methods how you generate the data in the materials and methods section.

Lines 312-313. I could not follow the logic here. Your data show correlation of PC and PS in HA samples, how could you jump to the conclusion that these two metabolites determine the distribution of *L. Tsingtauense*?

Line 323. Where are the microbial growth data?

Lines 362. Please rewrite the title.

Line 367. Is ample the right word?

Lines 373-374. Formatting errors.

Lines 387-389. Ascomycota consists of large number of fungal species including fungal pathogens. This statement and the rest of the paragraph are confusing and/or incorrect.

Response to Reviewer #1

1.L. 40-41 and 486-487: these are microbial taxa, not communities.

Response:

Thank you for pointing this out. We corrected 'microbial communities' with 'microbial taxa' in the revised manuscript. These revised items are marked with red fonts in line 41 and lines 343-344.

L. 41: '...Our results demonstrated a series of microbial taxa...'

L. 343-344: '...We also identified a series of microbial taxa...'

2. L. 54-56: I do not agree with this statement. There are plenty of studies on the microbiome associated to endangered plant species. Either provide an opportune reference to this statement or remove it.

Response:

Thank you for your suggestion. We revised this sentence and added the reference corresponding to the topic. These items are marked with red fonts in the revised manuscript in lines 55-57, and lines 474-476.

L. 55-57: '...Soil microorganisms are the basis for the sustainable survival of endangered plant populations (4). However, the symbiotic relationships between endangered plants and soil microorganisms have been poorly evaluated...'

L. 474-476: '(4) David, A.S., Quintana-Ascencio, P.F., Menges, E.S., et al., 2019. Soil microbiomes underlie population persistence of an endangered plant species. *Am Nat.* 194, 488-494. <https://doi.org/10.1086/704684>'

3. L. 56-57: this sentence is incomplete. It has no meaning.

Response:

Thank you for pointing this out. We removed this sentence to make the context consistency.

4. L. 120-122: either indicate the name of the primers or add opportune references.

Response:

We thank the reviewer for pointing this out. we added the name of the primers and references. These items are marked with red fonts in the revised manuscript in lines 384-389 and lines 644-650.

L. 384-389: '...The libraries of 16 S rDNA (bacteria) and internal transcribed spacer (ITS, fungi) were constructed by using the bacterial 16 S V3 +V4 primers (338F, 5'-ACTCCTACGGGAGGCAGCA-3'; 806R, 5'-GGACTACHVGGGTWTCTAAT-3') and the fungal ITS primers (ITS1F, 5'-CTTGGTCATTTAGAGGAAGTAA-3'; ITS2R, 5'-GCTGCGTTCTTCATCGATGC-3'), respectively (60, 61) ...'

L. 644-650: '(60) Song, Y., Li, X., Yao, S., et al., 2020. Correlations between soil metabolomics and bacterial community structures in the pepper rhizosphere under plastic greenhouse cultivation. *Sci Total Environ.* 728, 138439. <https://doi.org/10.1016/j.scitotenv.2020.138439>.

(61) Maldonado, J.E., Gaete, A., Mandakovic, D., et al., 2022. Partners to survive: *Hoffmannseggia*

doellii root-associated microbiome at the Atacama Desert. New Phytol. 234, 2126-2139. <https://doi.org/10.1111/nph.18080>.’

5. L. 151-154: for the network analysis, please indicate: correlation method used, statistical threshold considered as significant ($p < 0.05$? or < 0.01 ? or < 0.001 ?), algorithm used for network organization.

Response:

Thank you for your valuable suggestion. We added the correlation method and statistical analysis used in our revised manuscript. We also added Fig. S6 to support our findings. These changes are marked with red fonts in the revised manuscript in lines 160-166 and lines 418-419.

L. 159-165 ‘...The co-occurrence patterns of soil microbial communities at different altitudes (LA, MA, HA) were further explored. The modularity index in the six networks indicates that the generated network is modular. The links, nodes and average degree of the network of bacteria and fungi at HA are higher, which indicates that there is a more complex relationship between soil bacteria and fungi at LA. At the same time, Bacteroidota, Proteobacteria, Ascomycota and Basidiomycota were the most abundant at three altitudes (LA, MA, HA) and dominated in the network (Fig. S6).’

L. 418-419 ‘...The co-occurrence network of microbial phylum levels at all altitudes and different altitudes ($P < 0.05$, $r > 0.6$) was constructed by spearman test ...’

6. - L. 179-180: replace "attitude" with "altitude".

Response:

Thank you for pointing this out. We changed ‘attitude’ with ‘altitude’ with red fonts in our revised manuscript in line 101. In addition, we also checked the whole manuscript and revised 'alttitude' with 'altitude' in lines 86, 87, 99, 188, 373.

L. 101 ‘...than that at other altitudes LA and MA (Fig. 1C)...’

7. - Fig. 1B is not cited in the text.

Response:

We thank the reviewer for pointing this out. We revised Fig. 1A and 1B to make the sampling point site clear. We also cited Fig. 1B in the revised manuscript with red fonts in lines 98-100.

L. 98-100 ‘...According to the characteristics of *L. tsingtauense* community, we selected three types of soil sampling sites (high altitude, HA, medium altitude, MA, low altitude, LA) and recorded them (Fig. 1A and B) ...’

8. - L. 108 and L. 209: the statements about number of samples are discordant. Please, correct. I guess you have 18 samples and performed 36 sequencing analyses (one bacteria and one fungi per sample).

Response:

We appreciate your valuable comments. We have a total of 18 soil samples for bacterial and fungal determination. We corrected these relevant descriptions with red fonts in the revised manuscript in lines 118-119.

L. 118-119 ‘...we sequenced the amplicons of the bacterial 16S rRNA gene and fungal ITS region from 18 soil samples after mass filtration...’

9. - Fig. 3: move "unclassified bacteria" and "unclassified fungi" to the top of the cumulative bars, next to "unknown".

Response:

Thank you for pointing this out. We revised Fig. 3 according to your valuable suggestion.

10. - L. 249: write all these acronyms extensively the first time.

Response:

Thank you for pointing this out. We explained these acronyms at the first time. These items are marked with red fonts in our revised manuscript in lines 147-148.

L. 147-148 ‘...Average degree (AD), average pathway length (APL), modularity (MOD) and positive correlation line (PCL) were higher in the bacterial and fungal co-occurrence networks...’

11. - Supplementary tables S3-S6: why only the phylum is reported for the nodes?

Response:

Thank you for pointing this out. Phyla are very important and commonly used for classification of microorganisms in the studies of microbial communities, while sometimes is not reliable. We added more detailed annotation information of microorganisms in the construction of microbial topological roles. This relevant information is provided as Supplementary tables S3-S6 in the revised manuscript.

12. - L. 269-273: this information, currently placed in the legend of Fig. 5, must be reported in the materials and methods instead, and opportune references must be indicated to justify the choose of both Z_i and P_i thresholds.

Response:

Thank you for your valuable suggestion. We added the reference and detailed information of Z_i and P_i thresholds used in Fig. 5 in materials and methods section. These revised items are marked with red fonts in the revised manuscript in lines 421-423 and lines 670-674.

L. 421-423 ‘...The Z_i and P_i ($Z_i \geq 2.5$ or $P_i \geq 0.62$) of each node in the microbial symbiotic network were calculated according to the Guimerà (69, 70).’

L. 669-673 ‘(69) Guimerà, R., Amaral, L.A.N., 2005. Cartography of complex networks: modules and universal roles. *J. Stat. Mech.* 1742-1755. <https://doi.org/10.1088/1742-5468/2005/02/P02001>.

(70) Deng, Y., Jiang, Y., Yang, Y., et al., 2012. Molecular ecological network analyses. *BMC Bioinformatics.* 13, 113-133. <https://doi.org/10.1186/1471-2105-13-113>.’

13. - Fig. 7: why didn't you perform any statistics on these data? Why are these results not described anywhere in the text?

Response:

Thank you for pointing this out. We added the statistical analysis of the data used in Fig. 7. We also added the description of the Fig. 7 in lines 182-185 and lines 723-

725 in the revised manuscript. These revised items are marked with red fonts in the revised manuscript.

L. 182-185 ‘...We also predicted the functions of bacteria and fungi in RS. We found that ectomycorrhizal fungi (28.7%), endophytic fungi (8.3%), and N-fixing bacteria (8.1%) were more abundant and plant-pathogenic fungi (7.9%) were less abundant in RS at HA than in LA (Fig. 7) ...’

L. 723-725 ‘...Different letters indicate significant differences according to ANOVA followed by Tukey’s multiple comparison test ($P < 0.05$). Bars indicate standard errors ($n = 3$) ...’

14. - L. 325-340: I suggest to indicate the name of the metabolites instead of their codes, which are rather uninformative.

Response:

Thank you for your suggestion. We changed these codes with metabolites to provide effective information of *L. tsingtauense* rhizosphere microbial community. These revised items are marked with red fonts in lines 202-203, lines 204-206, lines 208-209, lines 215-216 in the revised manuscript.

L. 202-203 ‘...Spearman correlation analysis revealed that phosphatidylcholine, 15(S)-HETE, anemarsaponin E...’

L. 204-206 ‘...Additionally, phosphatidylcholine ($r = 0.72, 0.31, 0.87, \text{ and } 0.64$), 15(S)-HETE ($r = 0.68, 0.67, 0.63, \text{ and } 0.82$) ...’

L. 208-209 ‘...Anemarsaponin E ($r = -0.71, -0.45, -0.68, \text{ and } -0.80$) and BNR ($r = -0.83, -0.57, -0.84, \text{ and } -0.84$) were significantly negatively ...’

L. 215-216 ‘...phosphatidylcholine had the greatest effect on the distribution and growth of *L. tsingtauense*. Anemarsaponin E, 15(S)-HETE, phosphatidylcholine ...’

15. -L. 350: I guess you mean Table S7 instead of S8.

Response:

Thank you for pointing this out. We changed Table S8 with Table S7 in the revised manuscript. These revised items are marked with red fonts in line 741.

L. 741 ‘...The names of soil metabolites are shown in Table S7 ...’

16. - Conclusions: what you wrote is just a repetition of the study aim and findings. Try to re-write the conclusions highlighting the implications of the results and the future perspectives that they open.

Response:

Thank you for your suggestion. We rewritten the conclusions in the manuscript. Some duplications of study objectives and findings were removed and highlighted the implications of the results and the future perspectives in the conclusions. These revised items are marked with red fonts in lines 340-350.

L. 340-350 ‘This study is the first report of a comprehensive reference for understanding the rhizosphere microbiome of *L. tsingtauense*. We analyzed the characteristics and functions of soil microbial communities in *L. tsingtauense*, and determined the environmental factors affecting the differences in microbial communities. We also identified a series of microbial taxa, such as

Acidobacteriales, Xanthobacteraceae, Aspergillaceae, and Chaetomiaceae, and rhizosphere soil metabolites, such as phosphatidylcholine and phosphatidylserine, which play an essential role in the survival of *L. tsingtauense*. In sum, these results highlight the importance of microbial communities for *L. tsingtauense* growth and provides profound insights into the driving factors of microbial communities in ecosystems. At the same time, our results also provide a new perspective view for the protection of endangered plants.'

Response to Reviewer #2

1. Lines 54-57. Please rewrite these two sentences. For the first one, I would suggest the authors change the word "understudied" to other words such as "poorly evaluated".

Response:

Thank you for pointing this out. We revised these two sentences and changed "understudied" with "poorly evaluated". These revised items are marked with red fonts in lines 55-57 in the revised manuscript.

L. 55-57 '...Soil microorganisms are the basis for the sustainable survival of endangered plant populations (4). However, the symbiotic relationships between endangered plants and soil microorganisms have been poorly evaluated ...'

2. Lines 97-103. The methods to generate Fig 1 (a- f) data were missing. Can you mark your sampling sites in Fig 1a? I thought it is important to know where you collect your 3 bulk soil and 3 rhizosphere soil samples at each of the three altitude ranges (LA, MA, and HA). What criteria you used to select your sampling sites? Since you were examining altitudinal variation, how would you determine your sampling sites are good representation?

Response:

Thank you for your valuable suggestion. We added the methods to generate the data in Fig 1 (A-F) and sampling sites. We also revised Fig. 1A-B to make it clear and concise. For the criteria selection of sampling sites, we added the relevant description in materials and methods section in lines 353-358 and lines 360-368 with red fonts in the revised manuscript.

L. 353-358 'From March 2023 to November 2023, we investigated the wild distribution and characteristics of *L. tsingtauense* in mountain Lao, which is located in Qingdao, Shandong province of China (Fig. 1A). The growth slopes, slope directions and number of *L. tsingtauense* at different altitudes were recorded, respectively. To assess the growth status of *L. tsingtauense* influenced at different altitudes, we randomly selected 20 plants and recorded their heights.'

L. 360-368 'Three different representative altitude, which contains abundant wild *L. tsingtauense* community was used as the soil sample sites. These sampling sites included low altitude (486m, 120.6097°E, 36.2063°N), (516m, 120.6105°E, 36.2071°N), (524m, 120.6124°E, 36.2103°N), middle altitude (688m, 120.6082°E, 36.1786°N), (714m, 120.6104°E, 36.1893°N), (738m, 120.6091°E, 36.1916°N), and high altitude (975m, 120.6230°E, 36.1757°N), (985m, 120.6314°E, 36.1702°N), (1029m, 120.6277°E, 36.1739°N) and visualized by Arcmap 10.8. The sampling point of each altitude was in a 10 cm × 10 cm square area of three different sites, and the minimum distance between different sampling sites was above 50 m.'

3. Lines 117-128. Please briefly describe how you generate the sequencing libraries for both bacterial and fungal communities. The methods described here are confusing. What Illumina platform you used, Hiseq 2500 or Miseq?

Response:

Thank you for pointing this out. We used the Illumina MiSeq platform to analyze the bacterial and fungal communities. We removed the confusing part of the description method that generates sequencing libraries for bacterial and fungal communities. These revised items are marked with red fonts in lines 389-393 in the revised manuscript.

L. 389-393 ‘... PCR amplification was performed in a total volume of 50 μ L. The PCR amplification products were collected from a 2 % agarose gel and purified using Vazyme VAHTSTM DNA Clean Beads (Vazyme Biotech, Nanjing, China). The Illumina Mi Seq platform (Biomarker, Beijing, China) was used to perform high-throughput sequencing analysis of bacterial and fungal rRNA genes in the purified mixed samples.’

4. Line 135. I suspect the words "mass spectrometry" were missing after the QTOF?

Response:

Thank you for pointing this out. We added the words ‘mass spectrometry’ with red fonts in lines 399-400 in the revised manuscript.

L. 399-400 ‘...The metabolites were then detected using the Waters UPLC Xevo G2-XS QTOF mass spectrometry system ...’

5. Lines 152-153. Please rewrite the sentence. How about the negative correlations?

Response:

Thank you for your suggestion. We rewritten the sentence and added the negative correlation description. These revised items are marked with red fonts in lines 425-426 in the revised manuscript.

L. 425-426 ‘...All correlation analyses were performed using spearman ($r > 0$ indicates positive correlation, $r < 0$ indicates negative correlation) rank correlation analysis ...’

6. Line 177. As mentioned before, please provide the methods how you generate the data in the materials and methods section.

Response:

Thank you for your suggestion. We provided detailed methods for generating Fig.1 data in materials and methods section. These revised items are marked with red fonts in lines 353-358 in our revised manuscript.

L. 353-358 ‘From March 2023 to November 2023, we investigated the wild distribution and characteristics of *L. tsingtauense* in mountain Lao, which is located in Qingdao, Shandong province of China (Fig. 1A). The growth slopes, slope directions and number of *L. tsingtauense* at different altitudes were recorded, respectively. To assess the growth status of *L. tsingtauense* influenced at different altitudes, we randomly selected 20 plants and recorded their heights.’

7. Lines 312-313. I could not follow the logic here. Your data show correlation of PC and PS in HA samples, how could you jump to the conclusion that these two metabolites determine the distribution of *L. Tsingtauense*?

Response:

We thank you for pointing this out. We revised these description with red fonts in lines 198-199 in the revised manuscript.

L. 198-199 ‘...Volcanic maps indicated that there existed more phosphatidylcholine and phosphatidylserine in HA than in MA and LA (Figs. 8C–E) ...’

8. Line 323. Where are the microbial growth data?

Response:

Thank you for pointing this out. We revised the title with ‘Effects of environmental factors on *L. tsingtauense* and rhizosphere microbial community’ with red fonts in lines 200-201 in the revised manuscript.

L. 200-201 ‘Effects of environmental factors on *L. tsingtauense* and rhizosphere microbial community’

9. Lines 362. Please rewrite the title.

Response:

Thank you for your suggestion. We revise the title with ‘High altitude areas have more suitable soil environment for the growth of *L. tsingtauense*’ in lines 228 in the revised manuscript with red fonts.

L. 228 ‘High altitude areas might be the suitable soil environment for *L. tsingtauense*’

10. Line 367. Is ample the right word?

Response:

Thank you for pointing this out. We revised this sentence with red fonts in lines 232-233.

L. 232-233 ‘...the water available to plants in HA barren soils can support the normal physiological activities of the roots...’

11. Lines 373-374. Formatting errors.

Response:

Thank you for pointing this out. We changed the formatting to make it clear and concise. These revised items are marked with red fonts in lines 237-239 in the revised manuscript.

L. 237-239 ‘...Soil OM, TN, TP, and TK are critical nutrients essential for plant growth and development (19). The high TK content in HA areas aids the growth of *L. tsingtauense* by providing a source of essential nutrients ...’

12. Lines 387-389. Ascomycota consists of large number of fungal species including fungal pathogens. This statement and the rest of the paragraph are confusing and/or incorrect.

Response:

Thank you for pointing this out. We revised the description of some fungal species and their function. These revised items are marked with red fonts in lines 250-251 in the revised manuscript.

L. 250-251 ‘... Members of the Ascomycota are known to play a role in promoting plant development, resulting regulated carbon and N cycling...’

Re: Spectrum00966-24R1 (Altitudinal variation in rhizosphere microbial communities of the endangered plant *Lilium tsingtauense* and the environmental factors driving this variation)

Dear Dr. Xinqiang Jiang:

Your manuscript has been accepted, and I am forwarding it to the ASM production staff for publication. Your paper will first be checked to make sure all elements meet the technical requirements. ASM staff will contact you if anything needs to be revised before copyediting and production can begin. Otherwise, you will be notified when your proofs are ready to be viewed.

Sincerely,
Xia Ding
Editor
Microbiology Spectrum

Reviewer #1 (Comments for the Author):

The manuscript was revised according to my suggestions.

Reviewer #2 (Comments for the Author):

The authors have modified the manuscript following my previous suggestions.